# Patient-mix, programmatic characteristics, retention and predictors of attrition among patients starting antiretroviral therapy (ART) before and after the implementation of HIV "Treat All" in Zimbabwe

**Richard Makurumidze**[1,2,3]*, **Jozefien Buyze**[2], **Tom Decroo**[2,4], **Lutgarde Lynen**[2], **Madelon de Rooij**[1], **Trevor Mataranyika**[5], **Ngwarai Sithole**[5], **Kudakwashe C. Takarinda**[5,6], **Tsitsi Apollo**[5], **James Hakim**[1], **Wim Van Damme**[2,3], **Simbarashe Rusakaniko**[1]

1 College of Health Sciences, University of Zimbabwe, Harare, Zimbabwe, 2 Institute of Tropical Medicine, Antwerp, Belgium, 3 Gerontology, Faculty of Medicine & Pharmacy, Free University of Brussels (VUB), Brussels, Belgium, 4 Research Foundation of Flanders, Brussels, Belgiums, 5 AIDS & TB Unit, Ministry of Health & Child Care, Harare, Zimbabwe, 6 International Union Against Tuberculosis and Lung Disease, Paris, France

* rmakurumidze@ext.itg.be

## Abstract

### Background

Since the scale-up of the HIV "Treat All" recommendation, evidence on its real-world effect on predictors of attrition (either death or lost to follow-up) is lacking. We conducted a retrospective study using Zimbabwe ART program data to assess the association between "Treat All" and, patient-mix, programmatic characteristics, retention and predictors of attrition.

### Methods

We used patient-level data from the electronic patient monitoring system (ePMS) from the nine districts, which piloted the "Treat All" recommendation. We compared patient-mix, programme characteristics, retention and predictors of attrition (lost to follow-up, death or stopping ART) in two cohorts; before (April/May 2016) and after (January/February 2017) "Treat All". Retention was estimated using survival analysis. Predictors of attrition were determined using a multivariable Cox regression model. Interactions were used to assess the change in predictors of attrition before and after "Treat All".

### Results

We analysed 3787 patients, 1738 (45.9%) and 2049 (54.1%) started ART before and after "Treat All", respectively. The proportion of men was higher after "Treat All" (39.4.% vs 36.2%, p = 0.044). Same-day ART initiation was more frequent after "Treat All" (43.2% vs 16.4%; p<0.001) than before. Retention on ART was higher before "Treat All" (p<0.001).

**Data Availability Statement:** We have provided a de-identified dataset as a supplementary file (S1

Dataset). The following variables were removed to ensure confidentiality: district, province, health facility names, patient unique identification number and all dates (date of birth, testing, enrolment, ART initiation and follow-up visits).

**Funding:** Richard Makurumidze receives a PhD scholarship grant from the Institute of Tropical Medicine, funded by the Belgian Development Cooperation (DGD). The funders had no role in study design, data collection and analysis, decision to publish, or preparation of the manuscript.

**Competing interests:** The authors declare that they have no competing interests.

Among non-pregnant women and men, the adjusted hazard ratio (aHR) of attrition after compared to before "Treat All" was 1.73 (95%CI: 1.30–2.31). The observed hazard of attrition for women being pregnant at ART initiation decreased by 17% (aHR: $1.73 \times 0.48 = 0.83$) after "Treat All". Being male (vs female; aHR: 1.45; 95%CI: 1.12–1.87) and WHO Stage IV (vs WHO Stage I-III; aHR: 2.89; 95%CI: 1.16–7.11) predicted attrition both before and after "Treat All" implementation.

## Conclusion

Attrition was higher after "Treat All"; being male, WHO Stage 4, and pregnancy predicted attrition in both before and after Treat All. However, pregnancy became a less strong risk factor for attrition after "Treat All" implementation.

## Introduction

The highest number of people living with HIV (PLHIV) originate from the East and Southern African (ESA) region. However, this region has not yet met the Joint United Nations Programme for HIV/AIDS (UNAIDS) 90-90-90 targets, which were launched in 2014 to achieve epidemic control by 2020 [1]. By the end of 2019 in the ESA region, 87% of PLHIV knew their HIV status, with 72% of those diagnosed on antiretroviral therapy (ART) and only 65% of those on ART achieving sustained viral load suppression [2]. To close the gap, by the end of 2018, 93% of low- and middle-income countries and 100% of those designated as Fast Track countries had adopted the HIV "Treat All" policy [3].

Several clinical trials have shown the positive effects of "Treat All" on ART initiation, linkage to care, virologic suppression, and retention [4, 5]. Multiple studies have assessed the real-world effects of "Treat All" on ART initiation, linkage, virologic suppression, and retention. However, these studies have reported inconsistent results [6–13]. None of these previous studies has assessed if predictors for attrition changed since the transition to "Treat All". There is also lack of evidence outside of trial settings on the effects of "Treat All" on the patient mix, i.e. the diversity of patients in terms of diagnoses, disease severity, gender, age, socioeconomic status or functional status [14].

Zimbabwe has a generalised epidemic with an estimated HIV prevalence of 13.7% and 1.4 million PLHIV [15]. Of the estimated 1.4 million PLHIV, about 1.1 million are on ART. The country has progressed well towards meeting the 2020 UNAIDS 90-90-90 targets: 90% of PLHIV were aware of their HIV status; of that 86% were on ART and among those on ART, 73% were virally suppressed [2]. This remarkable progress towards the 90-90-90 targets has been mainly due to the adoption of the WHO 2015 HIV guidelines, which included the "Treat All" recommendation and differentiated care. The guidelines were launched on World AIDS Day, 1st of December 2016 and currently "Treat All" is being implemented throughout the country. Prior to the national scale-up, the "Treat All" recommendation was piloted in nine districts.

In Zimbabwe, as in many other low resource settings, whether there has been a change to the uptake of HIV care services, retention and viral suppression since the scale-up of "Treat All" is unknown. Moreover; men, adolescents and young adults remain underserved [16–19]. Whether the patient-mix has changed in favour of these previously underserved subgroups remains unclear. It is also not fully understood how programmatic procedures related to the provision of HIV care across the cascade of care have been adapted and their influence on ART outcomes.

Our study is the first to assess the performance of the ART programme across provinces in Zimbabwe since the start of HIV "Treat All" policy implementation and a follow-up on our previous national study on ART outcomes in the country [20]. In our earlier study, we found an increase in retention as compared to the prior 2011 evaluation. However, adolescents and young adults, patients with advanced HIV disease, receiving care at primary health care facilities and patients who started ART after 2013 when the country adopted the CD4 count cut-off of 500 cells per microlitre for ART initiation were at risk of attrition. In the current study, we assessed the association between "Treat All" and patient-mix, programmatic characteristics, retention and predictors of attrition by comparing two ART cohorts which started ART before and after the implementation of the "Treat All" policy.

## Methods and materials

### Study design

We conducted a retrospective cohort study using routinely collected individual patient-level data.

### Study setting

Since December 2016, Zimbabwe started scaling-up "Treat All" implementation. We used program data from the nine districts which piloted "Treat All" before it was expanded to the rest of the country. The nine districts that piloted "Treat All" were Chipinge, Bulilima, Gwanda, Harare, Mangwe, Makoni, Mazowe, Mutasa and Mutare. Zimbabwe is geographically divided into ten provinces and 63 districts. The nine pilot districts were selected from four provinces. The nine districts had a total of 385 health facilities. During the pilot period, 121 (31.4%) of them were being supported by the United States Presidential Emergency Plan for AIDS Relief (PEPFAR) through its two main implementing partners: International Training Education and Centre for Health (I-TECH) and Organisation for Public Health Intervention Development (OPHID). Of the nine districts, Harare and Mutare are mainly urban while the rest are mainly rural with a few urban centres. By the end of 2018, the nine districts accounted for about 25% of patients on ART in the country.

### Study population

Health facilities were included if they used an HIV programme electronic medical record, i.e. the electronic patient monitoring system (ePMS), which is being implemented in high-volume sites. Of 385 health facilities in the nine pilot districts, 131 (30%) had ePMS. Of 131 health facilities with ePMS, 72 had submitted their data to the national database at the end of December 2018 and were included in the study. The before "Treat All" cohort and the after "Treat All" cohort included all patients (children, adolescents, young adults, adults, elderly and pregnant) who were initiated on first-line ART between April-May 2016 (before "Treat All") and January-February 2017 (after "Treat All") in the nine districts. Patients who were restarted on ART after stopping treatment were excluded from the study.

### Data sources and study procedures

We used routinely collected National ART Program data in our study. Every patient within the program has a manual paper-based medical record, i.e. Patient ART/Opportunistic Infection Booklet. Patients medical records (baseline and routine visits) are first collected in this paper-based medical record. The data are then entered into the ePMS by either health care workers or data entry clerks available at some of the health facilities. Every month health

facilities submit an encrypted back-up of the ePMS data to the district-level where all the data are consolidated. The consolidated district-level data are then submitted to the nationallevel, where it is further consolidated into a one national database. The MoHCC, together with supporting partners, have put in place strategies to improve ePMS data quality and facility reporting rates. The strategies include regular support and supervision visits and on-site data verification. The ePMS database has also been designed with inbuilt check systems to improve data quality. Data for this study were extracted from the national database in an MS Excel format and exported to Stata were it was cleaned (synchronised and deduplicated) before analysis (S1 Fig).

## Study variables

The variables extracted for analysis from the ePMS included those to assess patient-mix, programmatic characteristics and retention. Patient-mix was defined as the diversity of patients in terms of age, sex, disease severity (WHO Stage, functional status), pregnancy and TB status at ART initiation. The age was categorised into children: 0–14 years, adolescents and young adults: 15–24 years, adults: 25–49 years and elderly: +50 years. Functional status was categorised as either impaired for patients who were bedridden and ambulatory, and normal for those working. The TB status at ART initiation was classified as negative on screening, presumptive or on TB treatment. Programmatic characteristics included HIV testing modality, level of care, ART regimen and time from HIV testing to ART initiation. Time from testing to ART initiation was defined as the time between the date of HIV testing and date of ART initiation. The patient follow-up status (active on treatment, lost to follow-up (LTFU), dead, transferred or stopped ART) and date ascertained were also extracted. LTFU patients were those whose last recorded clinic visit date, or pharmacy pill pick-up date, was ≥180 days before the date of data extraction from the ePMS. Patients were considered active on ART when their last recorded clinic visit date or pharmacy pill pick-up date was <180 days before the date of data extraction from the ePMS. The adverse outcome event was attrition, a composite variable which included those who died, LTFU and stopped ART.

## Data analysis

The data were analysed using Stata version 16.0 (Stata Corp, College Station, Texas, USA) [21]. Descriptive statistics (frequencies and proportions or medians and quartiles) were used to describe and compare patient-mix and programmatic characteristics (before and after "Treat All"). The Chi-square or Fisher's exact test was used to compare the proportions for the categorical variables before and after "Treat All". The Wilcoxon rank-sum test and t-test were used for skewed and symmetrical continuous variables, respectively.

We assessed the patient-mix by age (children, adolescents and young adults, adults and elderly), sex (males females), disease severity (WHO Stage, functional status), pregnancy and TB status at ART initiation by calculating the proportion initiated on ART for each category among all those who were initiated on ART for the before and after "Treat All" cohorts by dividing the number initiated on ART per category and the total initiated in either cohort. We assessed the change in programmatic characteristics (HIV testing modality, level of care, ART regimen and time from HIV testing to ART initiation) by calculating the proportion initiated on ART for each category among all those who were initiated on ART for the before and after "Treat All" cohorts by dividing the number initiated on ART per category and the total initiated in either cohort.

Time from testing to ART initiation was calculated by subtracting the date when the patient was initiated on ART with the date of HIV testing. The survival time was calculated as the time

from the date of ART initiation to the date of censoring or attrition. In the time-to-event analysis, patients classified as active on ART were censored on the date of their most recently recorded clinic visit or pharmacy pick-up while those who transferred out were censored on the date of transfer out. Kaplan Meier survival curves and statistics were used to estimate retention at 6, 12 and 24 months. The log-rank test was used to compare survival curves for different strata.

In the bivariate and multivariate analysis, we included variables with less than 30% missingness. Data on weight, height, CD4 and viral load were more than 50% missing and were excluded from the study. Functional status, WHO Stage, TB status at ART initiation and HIV testing modality were less than 30% missing and were included in the study after correcting for the missingness through multiple imputation. The data were assumed to be missing at random. Multiple imputation was conducted using multiple chains equations, and twenty imputed datasets were created and used for analysis. Cox proportional hazard models with a frailty variable for health facility were used to identify predictors of attrition. Proportional hazard assumptions were tested by comparing the observed with predicted survival curves and log-log plots. The cohort variable (before "Treat All", after "Treat All") was included in all bivariate and multivariate models. To control for the effect of disease progression on attrition, WHO stage and functional status were used as proxies since data on CD4 count were missing. A hierarchical approach was employed. All variables associated with p-value <0.1 in the bivariate analysis were included in the multivariate model. Sex was maintained in the model because of its clinical and programmatic relevance. Stepwise backward elimination was used until all variables in the model had a p-value < 0.05. To be able to evaluate whether the hazard of significant predictors changed between the before and after "Treat All" cohort we looked at the interactions between the significant variables and our primary exposure variable (i.e. cohort). They were assessed and added to the model, using the same hierarchical approach and stepwise backward elimination approach described above.

### Ethical (and regulatory) review

This study was submitted for ethical review and approval to the Institutional Review Board (IRB-1257/18) of the Institute of Tropical Medicine, Antwerp, Belgium; University of Zimbabwe Joint Research Ethics Committee (JREC/239/18) and Medical Research Council of Zimbabwe (MRCZ/A/2410). In addition, permission to conduct the study was sought from the Ministry of Health and Child Care (MoHCC). The data were anonymised by removing the district, province, health facility names, dates and patient unique identification number to maintain privacy and confidentiality.

## Results

### a) Study participants

We analysed a total of 3787 patients; 1738 (45.9%) from the before "Treat All" cohort and 2049 (54.1%) from the after "Treat All" cohort (Fig 1).

### b) Association between "Treat All" and patient-mix and programmatic characteristics

In the before and after "Treat All" cohort, the median age in years was 37 (interquartile range (IQR): 30–44) and 36 (IQR: 29–43), respectively. The proportion of men among those who started ART in the after "Treat All" cohort compared to the before "Treat All" cohort was 39.4% vs 36.2% (p = 0.044). There was no significant increase in the proportion of adolescents

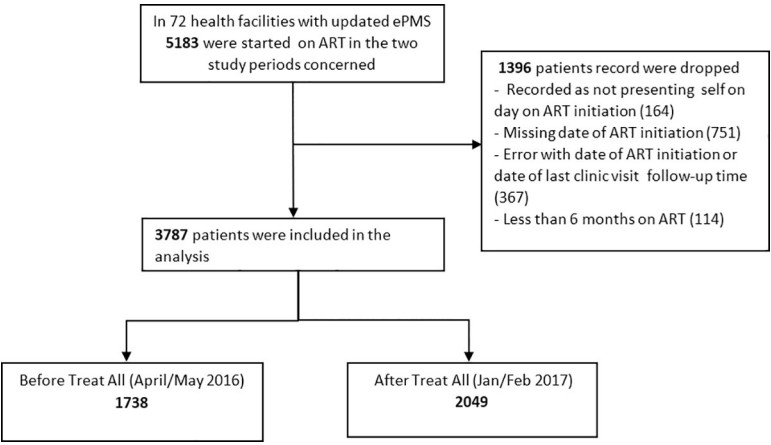

**Fig 1. Study participants.**

among those who started ART. The proportion of asymptomatic patients (WHO Stage 1) in the after "Treat All" cohort compared to the before "Treat All" cohort was 42.5% vs 26.1% (p <0.001) (Table 1).

The proportion of patients starting ART tested for HIV through voluntary counselling among all those initiated on ART in the after "Treat All" cohort compared to the before "Treat All" was 37.7% vs 28.6% (p <0.001) while the proportion tested in antenatal care services was 11.9% vs 17.6% (p <0.001). There was no significant change in the distribution of patients started on ART by the level of care. The median number in days between HIV diagnosis and ART initiation before and after "Treat All" were 21 (IQR: 5–95) and 1 (IQR: 0–24), respectively (S1 Table). The proportion of patients who started ART on the same day in the after compared to the before "Treat All" cohort was 43.2% vs 16.4% (p <0.001) (Table 2).

### c) Association between retention and "Treat All"

The maximum follow-up time for the before and after "Treat All" cohorts were 32.9 and 23.8 months while the median follow-up time were 19.7 (IQR, 6.1–28.1) and 16.0 (IQR, 8.9–19.7) months respectively. The total attrition for the before "Treat All" cohort was 142/1738 (8,2%; 9.9% stopped ART, 71.8% LTFU and 18.3% died) and after "Treat All" cohort was 165/2049 (8,1%; 23.0% stopped ART, 67.9% LTFU and 9% died) (S1 Table). Comparing the two cohorts, the 6 and 12 months retention for the before and after "Treat All" were 98.5% (95%CI; 97.8–99.0) and 95.1% (95%CI; 94.0–96.1) and, 97.0% (95%CI; 96.2–97.7) and 94.1% (95%CI; 92.9–95.1) respectively (Table 3). Retention during the first 12 months of treatment was higher before "Treat All" (log-rank, p<0.001) (Fig 2).

### d) Predictors of attrition before and after the implementation of "Treat All"

Among non-pregnant women and men, the adjusted hazard ratio (aHR) of attrition after "Treat All" compared to before "Treat All" was 1.73 (95%CI: 1.30–2.31). The observed hazard of attrition for women being pregnant at ART initiation decreased by 17% (aHR: 1.73*0.48 = 0.83) since "Treat All". Being male (vs female; aHR: 1.45; 95%CI: 1.12–1.87) and WHO Stage IV (vs WHO Stage I- III; aHR: 2.89; 95%CI: 1.16–7.11) predicted attrition both before and after "Treat All" implementation (Table 4).

**Table 1. Comparison of the patient-mix starting ART before (April-May 2016) and after (January-February 2017) the implementation of HIV "Treat All" in the 9 pilots districts in Zimbabwe.**

| Variables | Categories | Before "Treat All" | | After "Treat All" | | p-value[a] |
|---|---|---|---|---|---|---|
| | | n | (%) | n | (%) | |
| Total | | 1738 | (100) | 2049 | (100) | |
| Sex | Male | 629 | (36.2) | 807 | (39.4) | **0.044** |
| | Female | 1109 | (63.8) | 1242 | (60.6) | |
| Median age IQR($Q_1 - Q_3$) | | 37 (30–44) | | 36 (29–43) | | **0.008[b]** |
| Age groups | Children | 67 | (3.9) | 84 | (4.1) | |
| | Adolescents and young adults | 145 | (8.3) | 200 | (9.8) | 0.092 |
| | Adults | 1270 | (73.1) | 1513 | (73.8) | |
| | Elderly | 256 | (14.7) | 252 | (12.3) | |
| Functional Status | Bedridden | 1 | (0.1) | 3 | (0.1) | 0.220 |
| | Ambulatory | 148 | (8.5) | 146 | (7.1) | |
| | Working | 1588 | (91.4) | 1900 | (92.7) | |
| | Missing | 1 | (0.1) | 0 | (0.0) | |
| WHO Stage | WHO Clinical Stage 1 | 454 | (26.1) | 871 | (42.5) | **<0.001** |
| | WHO Clinical Stage 2 | 772 | (44.4) | 804 | (39.2) | |
| | WHO Clinical Stage 3 | 480 | (27.6) | 357 | (17.4) | |
| | WHO Clinical Stage 4 | 19 | (1.1) | 12 | (0.6) | |
| | Missing | 13 | (0.7) | 5 | (0.2) | |
| Pregnancy at ART initiation | Yes | 302 | (17.4) | 244 | (11.9) | **<0.001** |
| | [c]No | 1436 | (82.6) | 1805 | (88.1) | |
| TB Status at ART Initiation | Negative Screening | 1611 | (92.7) | 1908 | (93.1) | **0.023** |
| | TB Presumptive | 15 | (0.9) | 16 | (0.8) | |
| | On TB Treatment | 74 | (4.3) | 58 | (2.8) | |
| | Missing | 38 | (2.2) | 67 | (3.3) | |

[a]Chi-square test

[b]Wilcoxon rank-sum test

[c]Includes non-pregnant women and men, ART: antiretroviral therapy, TB: tuberculosis, IQR: interquartile range, WHO: World Health Organisation, Children: 0–14 years, Adolescents and young adults: 15–24 years, Adults: 25–49 years, Elderly: +50 years

The hazard ratio of attrition comparing pregnant women to non-pregnant women was significantly different (aHR: 0.48; 95%CI: 0.27–0.85) after "Treat All" compared to before "Treat All": the hazard ratio was 3.47 (95%CI: 2.36–5.11). The association of gender and WHO stage with attrition was not significantly different after "Treat All" compared to before "Treat All" (Table 4).

## Discussion

### Summary of main findings

The study is the first to assess the performance of the ART programme across provinces in Zimbabwe since the scale-up of "Treat All" policy. The study is a follow-up on previous national studies on ART outcomes in the country. The study is also among the first to assess if there has been a significant change in the predictors associated with attrition before and after the implementation of "Treat All". "Treat All" implementation resulted in lower retention and a higher risk of attrition. While the proportion of men among those who started on ART during "Treat All" increased, the proportion of adolescents and young adults among those starting ART did not significantly change between the two periods. After "Treat All", more patients

**Table 2. Comparison of programmatic characteristics before and after the implementation of HIV "Treat All" in the 9 pilots districts in Zimbabwe.**

| Variables | Categories | Before "Treat All" | | After "Treat All" | | [a]p-value |
|---|---|---|---|---|---|---|
| | | n | (%) | n | (%) | |
| Total | | 1738 | (100) | 2049 | (100) | |
| HIV testing modality | Antenatal | 306 | (17.6) | 243 | (11.9) | <**0.001** |
| | OI, TB & Illness | 580 | (33.4) | 634 | (30.9) | |
| | Occupational | 42 | (2.4) | 67 | (3.3) | |
| | Voluntary | 497 | (28.6) | 773 | (37.7) | |
| | Others | 63 | (3.6) | 52 | (2.5) | |
| | Missing | 250 | (14.4) | 280 | (13.7) | |
| Level of care | Clinic | 1143 | (65.8) | 1337 | (65.3) | 0.421 |
| | District/Mission Hospital | 460 | (26.5) | 571 | (27.9) | |
| | Provincial Hospital | 135 | (7.8) | 141 | (6.9) | |
| ART Regimen | TDF/3TC/EFV | 1646 | (94.7) | 1962 | (95.8) | 0.221 |
| | AZT/3TC/NVP(Children) | 43 | (2.5) | 41 | (2.0) | |
| | AZT/3TC/LPV/r(Children) | 19 | (1.1) | 17 | (0.8) | |
| | Others | 39 | (2,2) | 29 | (1.4) | |
| Time between HIV diagnosis and ART initiaton (days) | Same day | 285 | (16.4) | 885 | (43.2) | <**0.001** |
| | 1–14 days | 342 | (19.7) | 404 | (19.7) | |
| | 15–90 days | 501 | (28.8) | 283 | (13.8) | |
| | > 90 days | 388 | (22.3) | 268 | (13.1) | |
| | Missing | 222 | (12.8) | 209 | (10.2) | |

[a]Chi-square test, ART: Antiretroviral therapy, OI: opportunistic infection, TB: tuberculosis, TDF: tenofovir, AZT: zidovudine, 3TC: lamivudine, NVP: nevirapine, EFV: efavirenz, LPV/r: lopinavir/ritonavir

initiated ART on the same day as the day the HIV diagnosis was made. Being male, WHO Stage 4, and pregnancy predicted attrition in both before and after "Treat All". However, pregnant women at ART initiation though they were still at risk of attrition, the risk was lower after "Treat All" implementation than before.

## Association between "Treat All" and patient-mix and programmatic characteristics

In our study, we found the proportion of men in those starting ART under "Treat All" higher than before. A programmatic report from Lesotho also reported an increase in the proportion

**Table 3. Combined before and after HIV "Treat All" retention in care of patients who started antiretroviral therapy in the 9 pilot districts in Zimbabwe.**

| Cohort | Months since ART initiation | Retention (%) | 95% Confidence Interval |
|---|---|---|---|
| Combined | 6 | (97.7) | (97.1–98.1) |
| | 12 | (94.5) | (93.7–95.2) |
| | 24 | (88.1) | (86.6–89.5) |
| Before "Treat All" | 6 | (98.5) | (97.8–99.0) |
| | 12 | (95.1) | (94.0–96.1) |
| | 24 | (90.4) | (88.6–92.0) |
| After "Treat All" | 6 | (97.0) | (96.2–97.7) |
| | 12 | (94.1) | (92.9–95.1) |

After "Treat All": no patients had a follow-up of 24 months

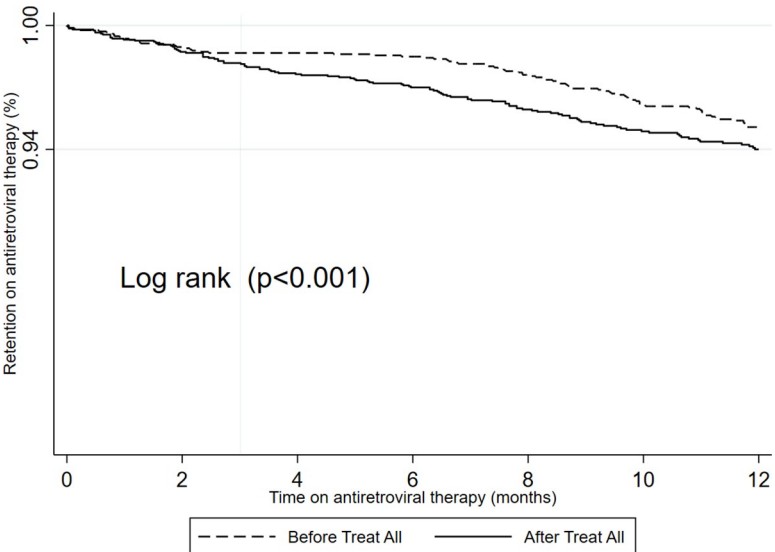

**Fig 2. Comparison of the retention at 12 months for patients who started before and after the implementation of HIV "Treat All" in the 9 pilots districts in Zimbabwe.**

of men among those starting ART under "Treat All" [22]. These findings are however contrary to an assessment which was conducted in 6 sub-Saharan African countries to assess the effects of HIV "Treat All" on rapid ART initiation, showing that men remain at risk of not initiating ART [23]. We believe that in our study the reason for a higher proportion of men among those initiating ART might have been due to demand creation, HIV testing and ART initiation strategies that were employed as part of "Treat All" implementation such as moonlight, index case testing and HIV self-testing [24–26]. The increase in patients testing through voluntary HIV testing services, as shown by this study, might also have benefited men.

We found that "Treat All" implementation did not increase the proportion of adolescents and young adults among those starting ART. To date, studies on whether HIV "Treat All" has improved ART initiation among adolescents are inconclusive. Positive effects have been shown in some studies, while others have shown that they are still at risk of not initiating ART [23, 27–29]. To our knowledge, no evidence-based strategies are currently being implemented in Zimbabwe on a broader scale under "Treat All" to promote ART initiation among adolescents and young adults. Strategies to improve ART initiation among adolescents which include multidisciplinary and adolescent-friendly HIV services together with peer counselling and support, should be tested [30, 31].

## Association between retention and "Treat All"

To date, the few studies assessing the real-life effects of "Treat All" have shown inconsistent findings regarding its impact on retention. Some studies have shown patients starting under "Treat All" likely to do better [11, 12], others worse [6, 7] while others showed no difference [8–10]. Several factors can explain the conflicting findings in these studies. These include different lengths of the follow-up period, settings, methods, study population and definition of outcomes. A systematic review and meta-analysis are needed to synthesise and consolidate the evidence.

In our study, patients who started ART during HIV "Treat All" had lower retention and a higher risk of attrition. This collided with findings from our previous evaluation that patients

**Table 4. Bivariate and multivariate analysis of factors associated with attrition for patients who started antiretroviral therapy before and after the implementation of HIV "Treat All" in the 9 pilots districts in Zimbabwe (Multiple imputation, N = 3787).**

| Variable | Categories | HR | p-value | (95%CI) | aHR | p-value | (95%CI) |
|---|---|---|---|---|---|---|---|
| Cohort | Before "Treat All" | 1 | | | | | |
| | After "Treat All" | **1.40** | **0.009** | **(1.09–1.80)** | **1.73** | **<0.001** | **(1.30–2.31)** |
| Sex | Female | 1 | | | | | |
| | Male | 1.12 | 0.336 | (0.89–1.41) | **1.45** | **0.005** | **(1.12–1.87)** |
| Age group | Adults | 1 | | | | | |
| | Children | 0.80 | **0.010*** | (0.42–1.52) | | | |
| | Adolescents and young adults | 1.30 | | (0.91–1.85) | | | |
| | Elderly | **0.55** | | **(0.37–0.84)** | | | |
| HIV testing modality | Voluntary | 1 | | | | | |
| | Antenatal | **1.52** | **0.002*** | **(1.10–2.11)** | | | |
| | Others | 0.84 | | (0.64–1.12) | | | |
| Baseline tuberculosis status | Negative screening | 1 | | | | | |
| | Presumptive tuberculosis | 2.05 | 0.233* | (0.83–5.06) | | | |
| | On tuberculosis treatment | 1.23 | | (0.70–2.16) | | | |
| WHO stage | I-III | 1 | | | | | |
| | IV | **2.52** | **0.0450** | **(1.02–6.21)** | **2.89** | **0.0220** | **(1.16–7.11)** |
| Functional status | Normal | 1 | | | | | |
| | Impaired | 1.06 | 0.830 | (0.62–1.83) | | | |
| Level of care | District/provincial hospital | 1 | | | | | |
| | Primary health facility | 0.81 | 0.448 | (0.47–1.39) | | | |
| Partner support | Not supported | 1 | | | | | |
| | Supported | 1.31 | 0.432 | (0.66–2.61) | | | |
| Pregnant when starting ART | [a] No | 1 | | | | | |
| | Yes | **2.08** | **<0.001** | **(1.58–2.75)** | **3.47** | **<0.001** | **(2.36–5.11)** |
| Pregnancy##Cohort | Not confirmed#After "Treat All" | | | | 1 | | |
| | Confirmed#After "Treat All" | | | | **0.48** | **0.0120** | **(0.27–0.85)** |

*Overall p-value

[a] Includes non-pregnant women and men, HR: hazard ratio, aHR: adjusted hazard ratio, CI: confidence interval, WHO: World Health Organisation, ART: antiretroviral therapy, #: interaction, Children: 0–14 years, Adolescents and young adults: 15–24 years, Adults: 25–49 years, Elderly: +50 years.

starting ART at a higher CD4 cut-off might be at risk of attrition [20]. Patients started on ART under "Treat All", possibly have low-risk perception due to the absence of symptoms and might be less motivated to adhere to lifelong daily ART [32, 33]. The other reason might be during the implementation of "Treat All" early on; more focus has been on ART initiation, including same-day ART initiation. In our study, we found that more patients are now being initiated on ART on the same day. Whether patients are adequately prepared, both psychological and clinically on the same day remains a critical question. The health care workers might be unable to prepare and provide adequate counselling on the importance of adhering to lifelong ART to these asymptomatic patients who might not appreciate its benefits hence defaulting treatment [38]. In a study conducted in Eswatini, the percentage of patients who never returned after the first same-day ART initiation visit doubled (from 3 to 6%) during the "Treat All" period [10]. In clinical trials, same-day ART initiation has shown either no difference or a positive effect on retention [34–37]. Few studies having been conducted to date to assess the real-world effects of same-day ART initiation [38], a call for further research.

The decrease in retention under "Treat All" has also been confirmed by routine programme reports from the national ART programme. LTFU continues to be a challenge and accounts for most of the decrease in retention. This is thought to have an administrative reason [20], and MoHCC and the supporting implementing partners have since instituted a National ART Census Survey to look into that. This, therefore, calls for the country to develop and implement innovative, evidence-based strategies that have been shown to improve retention. The process has already started with differentiated ART delivery models (Community ART refills groups-CARGS, adherence clubs, fast track and outreach) currently being scaled-up. By the end of 2018, around 30% of stable adolescents and adults on ART were receiving ART through at least one differentiated ART delivery model [39]. Systems to track and trace defaulters have also been put in place.

## Predictors of attrition before and after the implementation of "Treat All"

In our study, we found pregnant women at ART initiation still at risk of attrition though the risk was lower than before "Treat All", and this finding was a positive development. We hypothesise that the reduction might be due to the expansion of "Treat All" initiative to the rest of the PLHIV. For a long period, "Treat All" has been specifically focused on the subgroup of HIV positive pregnant women and delivered as Option B+, which might have created some "stigma" [40, 41]. Now, "Treat All" is applied to all patients; this label might have been removed. Current efforts to retain pregnant and breastfeeding women on ART which include the integration of services, family-centred approaches, and the use of lay healthcare providers which include mentor mothers should be optimised and continued [42, 43]. Documentation and follow-up of HIV positive pregnant women with complications referred from lower to higher-level health facilities (maternal waiting homes) should also be improved so that these referrals are not treated as LTFU.

In the current study, adolescents and young adults had comparable retention with adults contrary to other studies. The finding is contrary to other programmatic HIV "Treat All" studies to date [6, 7, 9–11]. Zimbabwe has employed strategies to retain adolescents in care, mainly through its home-grown community adolescent's treatment supporters (CATS) model. Evidence has shown the CATS model to be effective in retaining adolescents in HIV care [44]. To date, the model has been scaled-up to most of the districts. Resources to fully implement the CATS model throughout the country should be mobilised.

We found men to be at risk of attrition under "Treat All", a finding similar to other studies [7, 9]. Differentiated service delivery strategies that have been shown to retain men in ART care under HIV "Treat All" which includes tailored awareness on the health benefits of early ART start, accelerated linkage to care, decreasing logistical barriers to HIV care, patient-centred approach, flexible clinic hours and tracking and tracing of those missing appointments should be strengthened, maintained and optimised [45].

## Strength and limitations

Compared to other implementation studies conducted to date comparing outcomes before and after "Treat All" [6–13], our study sample size was large. The pilot districts were distributed across all the regions of the country. The study was the first to assess the performance of the ART programme across provinces since the start of HIV "Treat All" policy implementation and allowed us to confirm findings from our previous evaluation. However, our study had limitations. We evaluated the effect of HIV "Treat All", considering it as one intervention, though it has several components across the cascade of care. We were unable to disentangle these components and assess their individual effects on attrition. Our study was based on health facilities

with ePMS, which is only available in high volume health facilities, and this might affect the generalisability. However, sites with ePMS contain most patients on ART in the country. We could also not assess re-engagement in care since the ePMS database only reports the most recent status of the patient with no record of previous disengagements. Most of the attrition in our study was explained by LTFU. Some of these patients reported as LTFU might be still alive and on ART (unreported self-transfers), stopped ART or unreported deaths [46].

In our study, we assessed the change in the patient-mix before and after "Treat All" and with the assumption that if a previously underserved subgroup is proportionally more frequent under "Treat All" this might imply that uptake has increased in this group. A more direct approach used to measure uptake relies on assessing the fraction of all HIV-infected individuals or the fraction of all ART-eligible individuals, who are initiated on ART within a specified follow-up period from the date of HIV testing [47]. We were unable to assess the fraction initiated on ART within a specified period because ePMS does not collect individual patient-level data on those who do not start ART among all those tested for HIV. However, the ePMS has been updated to include an HIV testing module which collects individual patient-level testing and linkage to ART data.

Because we used routine electronic programme data collected retrospectively, we had to deal with missing data through multiple imputation. Multiple imputation involves strong untestable assumptions [48]. Data on weight, height, CD4 and viral load were more than 50% missing, so could not be assessed as confounders. To assess the effect of disease progression on attrition, we then used WHO staging and functional status as proxies in our model. Because of missing data, we were also unable to compare the viral load suppression rates between the before and after HIV "Treat All" cohorts. Unfortunately, it was impossible to solve this by merging ePMS and laboratory data, due to missing or inconsistent format of the unique patient identifier between the various viral load testing laboratories. Strategies to improve data quality and making sure the existing electronic data collection systems are interoperable need urgent attention.

This observational study looked into the association of individual patient and programme characteristics and retention in care. We are aware that lumping patients from all facilities together may have caused a loss of information on specific health facility-level factors that influence retention, and that could not be studied.

### Future research

Considering the inconsistency of evidence in the programmatic effects of "Treat All", there is a need for more research in the domain. We evaluated the effect of "Treat All" as one intervention but "Treat All" is being implemented as a package with several components across the cascade care (HIV testing, linkage to care, clinical and psychological readiness assessment, retesting before ART initiation, same-day ART initiation, follow-up and retention). Our study was unable to unpack these components and assess their individual effects on retention in care. On top of the HIV cascade of care issues; there are also health systems challenges anticipated from the abrupt increase in the number of patients now being followed-up in care. Further research should explore how these components separately, e.g. same-day ART initiation and health system changes affect retention in care.

Men continue to be at risk of attrition under "Treat All"; further research is needed on how ART delivery can be streamlined to retain them in care. Most studies, including ours, have focused on patient-level factors on ART outcomes. Lumping patients from different facilities together may cause loss of information on specific health facility-level factors that influence retention. This, therefore, calls for studies looking into the impact of health facility-level

characteristics on ART outcomes since performance might vary across facilities. Finally, as more studies looking into the real-world effects of "Treat All" surface, there is a need for systematic reviews and meta-analyses summarising outcomes across the cascade of care and evaluation of strategies to optimise "Treat All" implementation.

## Conclusion

In conclusion, our study found that since the implementation of "Treat All", retention in care was lower and the risk of attrition was higher. Male sex, advanced HIV disease and pregnancy were risk factors for attrition both before and during the implementation of HIV "Treat All". However, pregnant women at ART initiation though they were still at risk of attrition, the risk was lower after "Treat All" implementation than before. Strategies to retain men and to improve ART initiation among adolescents and young adults should be prioritised. Further research should explore how different components of HIV "Treat All" implementation (HIV testing, linkage, same-day ART initiation, clinical and psychological preparation) and health system/facility issues have a bearing on patient outcomes.

## Supporting information

**S1 Fig. Data management system for the National ART Program in Zimbabwe.**
(TIF)

**S1 Table. Outcomes, follow time and time from testing to ART initiation for patients who started antiretroviral therapy before and after the implementation of HIV "Treat All" in the 9 pilots districts in Zimbabwe.**
(DOCX)

**S2 Table. Attrition for patients who started antiretroviral therapy before and after the implementation of HIV "Treat All" in the 9 pilots districts in Zimbabwe.**
(DOCX)

**S1 File. Retention and predictors of attrition among patients who started antiretroviral therapy in Zimbabwe's national antiretroviral therapy programme between 2012 and 2015.** PLoS ONE 15(1): e0222309. https://doi.org/10.1371/journal.pone.022230.
(PDF)

**S1 Dataset. Dataset for patients who started antiretroviral therapy before and after the implementation of HIV "Treat All" in the 9 pilots districts in Zimbabwe.**
(DTA)

## Acknowledgments

The authors would like to thank the following organisations for support during the study—Ministry of Health and Child Care (MoHCC); the University of Zimbabwe, College of Health Sciences (UZCHS), Department of Community Medicine; Institute of Tropical Medicine (ITM) in Antwerp, Belgium; the University of Zimbabwe, Zimbabwe AIDS Prevention Project (ZAPP-UZ); PEPFAR Zimbabwe and its main implementing partners, I-TECH and OPHID for supporting the HIV "Treat All" pilot. The authors are grateful to the team that assisted in ePMS data management prior to analysis. The authors also appreciate various stakeholders and implementing partners who have been supporting the MoHCC towards improving the MoHCC AIDS and TB Program data quality through various initiatives, including supporting the ePMS. Of note, is the Global Fund for AIDS, Tuberculosis, and Malaria (GFATM) which

supported MoHCC in setting up the ePMS. Finally, we thank the front line health workers and PLHIV on ART whose records were used.

## Author Contributions

**Conceptualization:** Richard Makurumidze, Jozefien Buyze, Tom Decroo, Lutgarde Lynen, Madelon de Rooij, Trevor Mataranyika, Ngwarai Sithole, Kudakwashe C. Takarinda, Tsitsi Apollo, James Hakim, Wim Van Damme, Simbarashe Rusakaniko.

**Data curation:** Richard Makurumidze, Jozefien Buyze, Trevor Mataranyika.

**Formal analysis:** Richard Makurumidze, Jozefien Buyze.

**Funding acquisition:** Lutgarde Lynen.

**Investigation:** Richard Makurumidze.

**Methodology:** Richard Makurumidze, Jozefien Buyze, Tom Decroo, Lutgarde Lynen, Madelon de Rooij, Trevor Mataranyika, Ngwarai Sithole, Kudakwashe C. Takarinda, Tsitsi Apollo, James Hakim, Wim Van Damme, Simbarashe Rusakaniko.

**Resources:** Lutgarde Lynen.

**Supervision:** Lutgarde Lynen, Simbarashe Rusakaniko.

**Validation:** Jozefien Buyze.

**Writing – original draft:** Richard Makurumidze.

**Writing – review & editing:** Jozefien Buyze, Tom Decroo, Lutgarde Lynen, Madelon de Rooij, Trevor Mataranyika, Ngwarai Sithole, Kudakwashe C. Takarinda, Tsitsi Apollo, James Hakim, Wim Van Damme, Simbarashe Rusakaniko.

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
