## [Decision Letter · Decision Letter 0]

11 Aug 2020

PONE-D-20-20707

Patient and programmatic characteristics, retention and predictors of attrition among patients starting antiretroviral therapy (ART) before and after the implementation of HIV “Treat All” in Zimbabwe

PLOS ONE

Dear Dr. Makurumidze,

Thank you for submitting your manuscript to PLOS ONE. After careful consideration, we feel that it has merit but does not fully meet PLOS ONE’s publication criteria as it currently stands. Therefore, we invite you to submit a revised version of the manuscript that addresses the points raised during the review process.

We look forward to receiving your revised manuscript.

Kind regards,

Professor Kwasi Torpey, MD PhD MPH

Academic Editor

PLOS ONE

Journal Requirements:

3. Please ensure that you refer to Figure 1 in your text as, if accepted, production will need this reference to link the reader to the figure.

4. We note you have included a table to which you do not refer in the text of your manuscript. Please ensure that you refer to Table 4 in your text; if accepted, production will need this reference to link the reader to the Table.

Reviewers' comments:

Reviewer's Responses to Questions

**Comments to the Author**

1. Is the manuscript technically sound, and do the data support the conclusions?

Reviewer #1: Yes

Reviewer #2: Partly

2. Has the statistical analysis been performed appropriately and rigorously? 

Reviewer #1: Yes

Reviewer #2: I Don't Know

3. Have the authors made all data underlying the findings in their manuscript fully available?

Reviewer #1: No

Reviewer #2: Yes

4. Is the manuscript presented in an intelligible fashion and written in standard English?

Reviewer #1: Yes

Reviewer #2: No

5. Review Comments to the Author

Reviewer #1: • The manuscript appears to be fairly sound with the authors having captured the critical data in line with the title. The language used is clear, easy to understand and unambiguous. The manuscript is informative and adds to the technical knowledge and understanding of subject area.

• The methods are sufficiently detailed.

• The study design is in sync with the subject of interest, and the study has sound statistical analyses. However, data is not accessible thus a limitation. In addition, the study setting information is not detailed enough for example, what percentage of clients do these nine districts serve?

• The results are well outlined with tables having correct titles

• The discussion section is very strong with relevant citations made

• The conclusions are solid and relating well with the findings.

• The authors should clarify the following sections to avoid confusion

• In background, improve on the country and specific region context and literature review on the subject matter

• Under results, the authors were also assessing uptake of test and start among the different populations (men, adolescents) which is not related to the manuscript subject matter

• The age cut offs for the four categories i.e children, adolescents and young adults, adults and elderly not clear

• In the discussion, the authors aim to demonstrate low uptake of test and start among adolescents and increased uptake among men, however, the methods and data as shared does not fully support this

• The authors introduced ART initiation in the discussion which was not in the study design nor population nor area of focus “We found that “Treat All” implementation did not increase the proportion of 317 adolescents and young adults among those starting ART. To date, studies on whether 318 HIV “Treat All” increases rapid ART initiation among adolescents are inconclusive”.

Reviewer #2: I request that the editor gets a biostatistician to provide expert advise on the analysis section please.

The authors do not clearly state their aims or objectives in the manuscript. Thus it is difficult to follow through to assess claims or research questions being proposed by the authors as well as the findings that support them.

There are a lot of variables being assessed with no clear outcomes of interest.

The structure of most of the sentences needs re-phrasing for a clear understanding.

The methods section is not well written and needs to be detailed but clear and concise. There were no definitions of variables or outcomes. The study procedure needs to be written with more details so the study can be replicated elsewhere by following the detailed study procedure description by authors.

The methods section in the current state that it is does not allow other investigators elsewhere to follow and fully replicate the study.

Most comments can also be found in the attachment that has been uploaded

6. PLOS authors have the option to publish the peer review history of their article (what does this mean?). If published, this will include your full peer review and any attached files.

Reviewer #1: No

Reviewer #2: No

---

## [Author Response · Author response to Decision Letter 0]

30 Sep 2020

We have submitted our comments the attached document - Response to reviewers comments.

---

## [Editor Report · Decision Letter 1]

5 Oct 2020

Patient-mix and programmatic characteristics, retention and predictors of attrition among patients starting antiretroviral therapy (ART) before and after the implementation of HIV “Treat All” in Zimbabwe

PONE-D-20-20707R1

Dear Dr. Makurumidze,

We’re pleased to inform you that your manuscript has been judged scientifically suitable for publication and will be formally accepted for publication once it meets all outstanding technical requirements.

Kind regards,

Professor Kwasi Torpey, MD PhD MPH

Academic Editor

PLOS ONE

Additional Editor Comments (optional):

Comments addressed
---

## [Editor Report · Acceptance letter]

9 Oct 2020

PONE-D-20-20707R1 

Patient-mix, programmatic characteristics, retention and predictors of attrition among patients starting antiretroviral therapy (ART) before and after the implementation of HIV “Treat All” in Zimbabwe 

Dear Dr. Makurumidze:

I'm pleased to inform you that your manuscript has been deemed suitable for publication in PLOS ONE. Congratulations! Your manuscript is now with our production department. 

Kind regards, 

on behalf of

Professor Kwasi Torpey 

Academic Editor

PLOS ONE